# Effects of Compostable Packaging and Perforation Rates on Cucumber Quality during Extended Shelf Life and Simulated Farm-to-Fork Supply-Chain Conditions

**DOI:** 10.3390/foods10020471

**Published:** 2021-02-20

**Authors:** Abiola Owoyemi, Ron Porat, Victor Rodov

**Affiliations:** 1Department of Postharvest Science, ARO—the Volcani Center, P.O. Box 15159, Rishon LeZion 7505101, Israel; abiola.owoyemi@mail.huji.ac.il (A.O.); Vrodov@volcani.agri.gov.il (V.R.); 2The Robert H Smith Faculty of Agriculture, Food and Environment, The Hebrew University of Jerusalem, Rehovot 76100, Israel

**Keywords:** cucumber, modified atmosphere, postharvest, plastic, compostable

## Abstract

Cucumbers are highly perishable and suffer from moisture loss, shriveling, yellowing, peel damage, and decay. Plastic packaging helps to preserve cucumber quality, but harms the environment. We examined the use of compostable modified atmosphere packaging (MAP) with different perforation rates as a possible replacement for conventional plastic packaging materials. The results indicate that all of the tested types of packaging reduced cucumber weight loss and shriveling. However, compostable MAP with micro-perforations that created a modified atmosphere of between 16–18% O_2_ and 3–5% CO_2_ most effectively preserved cucumber quality, as demonstrated by reduced peel pitting, the reduced appearance of warts and the inhibition of yellowing and decay development. Overall, micro-perforated compostable packaging extended the storage life of cucumbers under both extended shelf conditions and simulated farm-to-fork supply-chain conditions and thus may serve as a replacement for the plastic packaging currently used to preserve the postharvest quality of cucumbers.

## 1. Introduction

Fresh fruit and vegetables (F&V) are living organisms and, as such, are very perishable food items with relatively short postharvest storage lives [1]. According to the United Nations (UN) Food and Agriculture Organization (FAO) report, about one-third of all food produced on the planet gets lost and is not consumed by humans [2]. The F&V category accounts for 44% of total global food losses [2,3]. Furthermore, the FAO reported that between 45 and 55% of all F&V produced worldwide is lost or wasted along the supply chain [2]. Most food losses in medium- and high-income countries occur at the retail and consumption stages of the food supply chain [2,4]. According to a U.S. Department of Agriculture Economic Research Service (USDA-ERC) report, in the U.S. alone, fruit and vegetables losses at the retail and consumption stages are estimated at 18.4 and 25.2 billion pounds, respectively, which represent losses of 28% of all fruit and 30% of all vegetables at these two stages [5].

Retail packaging is one of the key strategies for preserving food freshness and quality and reducing food losses [6,7,8]. A recent study by the American Institute for Packaging and the Environment proposed that proper use of packaging might reduce 10–15% of food waste at the store level and 20–25% of food waste at the household level [9]. Besides its basic role of containing the food, a retail F&V package may create an optimized modified-atmosphere and modified-humidity conditions to retain freshness, slow down ripening and senescence processes, and reduce the development of decay and physiological disorders [10,11]. The creation of an optimal atmosphere and optimal humidity conditions within the retail packaging greatly depends on the respiration rate of the produce and storage temperatures, as well as on the gas permeability and perforation rates of the packaging. Macro-perforations (4- to 8-mm holes) allow for the free exchange of gases through the package and so do not support the creation of a modified atmosphere. In contrast, in sealed packages or packages with micro-perforations, which are typically 30–350 µm in size, we may find a modified gaseous composition within the package, which will vary with the amount and type of packed produce and storage temperatures [12,13].

Despite the great advantages of plastic packaging for preserving freshness and reducing food losses, today, its use has negative environmental connotations and encounters an unfavorable public attitude. Currently, most plastic packages are made of conventional polymers that are responsible for environmental pollution due to their very slow degradation kinetics [14]. To address the issue of plastic pollution, more than 400 organizations have recently signed the New Plastics Economy Global Commitment, which endorses a common circular economy vision according to which, by 2025, all plastic packaging will be reusable, recyclable or compostable [15]. The use of compostable polymers may provide a promising sustainable solution, provided that the packaging made from those polymers extends produce storage life at least as effectively as conventional plastic packaging [16,17,18].

In the current study, we examined the effects of compostable packages with different perforation rates on the quality of fresh cucumber fruits over time under different storage conditions. Cucumbers are highly perishable and are very susceptible to moisture loss, shriveling, yellowing and the development of physiological injuries, and microbial spoilage [19]. It was previously reported that conventional plastic modified-atmosphere packaging (MAP) effectively extends the storage life of cucumbers and alleviates the development of chilling injury in cucumbers [20,21,22]. Suslow and Cantwell [19] reported that cucumbers might tolerate relatively low O_2_ levels of 3–5%, but will not tolerate more than 10% CO_2_, and Manjunatha and Anurag [21] reported that the optimal gas conditions for MAP for cucumbers are between 12 and 17% O_2_, and 5 and 10% CO_2_. In this study, we examined the efficacy of compostable MAP for preserving cucumber quality and the use of such packaging as a possible replacement for the conventional plastic retail packaging currently in use.

## 2. Materials and Methods

### 2.1. Plant Material

Cucumbers (Beit Alpha type) were harvested on 31 December 2019 from a commercial greenhouse in Ahituv, Israel, and brought within 1 h to the Department of Postharvest Science, Agricultural Research Prganization (ARO), The Volcani Center. There, the fruits were divided into 90 groups, with each group containing eight fruit that had a uniform green color. The fruit were distributed into five different packaging treatments as described below. Each treatment included 18 packages, with three packages for each evaluation point.

### 2.2. Packaging Treatments

The experiment included five treatments: (1) control, (2) non-perforated compostable package, (3) micro-perforated compostable package, (4) macro-perforated compostable package, and (5) commercial macro-perforated polypropylene package. The packages made of a compostable polyester blend were 35 µm thick and 30 × 40 cm in size and were supplied by TIPA^®^ Corp. (Hod HaSharon, Israel). The polypropylene packages of similar thickness and dimensions were supplied by R.O.P. Ltd. (Hahotrim, Israel). The oxygen transmission rates (OTR) of the compostable and polypropylene packages were 875 and 2019 cc/m^2^/day, respectively; and the water vapor transmission rates (WVTR) of the compostable and polypropylene packages were 40.8 and 8.1 gr/m^2^/day, respectively.

Micro-perforations were made by making 8 holes with a 0.5-mm needle (PIC Ago Ipodermico, 25G hypodermic needle, Grandate, Italy) and macro-perforations were made by making 8 holes (6 mm in width) with a hole puncher commonly used to punch holes in paper. The bags were sealed using a manual impulse heat sealer (Swery Electronics Ltd., Petah Tikva, Israel). Control, non-bagged fruits were kept in open rigid polyethylene terephthalate (PET) containers (Plasto-Vack, Rishon LeZion, Israel). The headspace oxygen and carbon dioxide concentrations in the packages were measured with an OxyBABY gas analyzer (WITT Gasetechnik GmbH and Co KG, Witten, Germany).

### 2.3. Storage Conditions

The fruit were stored under two different regimes. The first regime included continuous storage at 22 °C, to simulate the extended marketing period on a non-refrigerated shelf (“Extended shelf life”). The second regime simulated the farm-to-fork supply chain, including 2 days of distribution at 15 °C + 2 days of shelf life at 22 °C + 1 or 2 weeks of home storage at 4 °C in a 600-L home refrigerator (“Supply chain + home storage”). The relative humidity (RH) at the different storage rooms at 22, 15, and 4 °C was ~60%, ~75 and ~92%, respectively.

### 2.4. Quality Evaluations

Quality evaluations were conducted at time zero and after 5, 10, and 15 days under shelf conditions (22 °C) and after exposure to simulated distribution and marketing conditions (2 days at 15°C + 2 days at 22 °C) and 1 and 2 additional weeks of refrigerated home storage at 4 °C.

Quality evaluations included measurements of fruit weight loss (percentage of initial weight) and visual estimations of water condensation on the packages, shriveling, peel pitting, and the appearance of warts (0 = none, 1 = slight, 2 = moderate, and 3 = severe). Peel color was evaluated according to a visual yellowing scale (0 = dark green, 1 = light green, 2 = green–yellow, 3 = mostly yellow). The incidence of microbial decay was measured as the percentage of infected fruit. Flavor was rated on a 9-grade hedonic scale, in which 1 = very bad and 9 = excellent. Finally, an overall visual-acceptance score was determined according to a 5-grade scale, in which 1 = very bad, 2 = poor, 3 = fair, 4 = good, and 5 = excellent. A score of 2.5 was defined as the minimum acceptability threshold. All visual and sensory evaluation scores were assigned by three trained panelists.

### 2.5. Statistical Analysis

One-way analysis of variance (ANOVA) and Tukey’s honestly significant difference (HSD) pairwise comparison tests were applied using the JMP statistical software program, version 7 (SAS Institute Inc., Cary, NC, USA). Microsoft Office Excel was used to calculate means, standard deviations and standard errors.

## 3. Results

### 3.1. Visual Appearance

At time zero, the fruits were firm, had a dark green color and were free of blemishes (Figure 1). The polypropylene packages were clear and transparent, whereas the compostable packages were slightly hazy (Figure 1).

During the extended shelf-conditions at 22 °C, the quality of the control, non-packed cucumbers deteriorated quickly (Figure 2). After 10 days of shelf conditions, these fruits were shriveled and had a light green–yellowish color. In contrast, the cucumbers in all of the perforated packages remained smooth, green, and healthy, while those in the non-perforated packages suffered from decay (Figure 2). After 15 days of shelf conditions, the control fruit were severely shriveled and turned yellow–brown. The cucumbers in the macro-perforated packages, both the compostable ones and those made of polypropylene film, senesced and turned yellow; the cucumbers in the non-perforated compostable packages suffered from senescence and decay (Figure 2). At that time, only the cucumbers in the micro-perforated packages remained smooth and green.

Quality loss was also evident among the control fruits during the supply-chain simulation and subsequent refrigerated home storage (Figure 3). After 1 week of refrigerated home storage, the control fruits were shriveled, but remained green; whereas the packaged fruits were still smooth and attractive. After 2 weeks of home storage, the control fruits were severely shriveled and had turned light green–yellowish; whereas the quality of the cucumbers packed in the micro-perforated compostable package remained adequate. The cucumbers in the macro-perforated packages were of medium quality and the cucumbers in the non-perforated packages were slimy and macerated due to bacterial spoilage (Figure 3).

### 3.2. Atmosphere Compositions

After 5 days of shelf life at 22 °C, as well as after the supply-chain treatment (2 days at 15 °C + 2 days at 22 °C), the O_2_ levels in the non-perforated compostable packages were extremely low (nearly 1.5%) and the CO_2_ levels in those packages had increased to nearly 7%. Later on, the O_2_ levels in the sealed compostable packages gradually increased to 16–17% and the CO_2_ levels gradually decreased to 4–5% (Figure 4). In the micro-perforated compostable packages, the O_2_ levels decreased moderately to just 15–17%, while the CO_2_ levels increased moderately to 2–5% (Figure 4). The O_2_ and CO_2_ levels in the macro-perforated packages were similar to those detected in regular air (Figure 4).

### 3.3. Water Loss and Condensation

A major problem with cucumber preservation is extensive water loss after harvest. Our results indicated that non-bagged, control fruits lost up to 39% of their initial weight after 15 days of storage under shelf conditions (22 °C) and up to 25% of their initial weight after storage under the simulated supply-chain conditions and 2 weeks of refrigerated home storage (Figure 5a,b). In contrast, the cucumbers in the various packages had significantly (*p* ≤ 0.05) lower weight loss, and lost only 13–20% of their initial weight under the shelf conditions and 5–9% of their initial weight after the supply-chain treatment and two additional weeks of refrigerated home storage (Figure 5a,b).

Condensation was minimal in all of the packages kept under shelf conditions, but reached slight and moderate levels in the polypropylene packages after 1 and 2 weeks of refrigerated home storage, respectively. The condensation levels in all of the compostable packages were significantly (*p* ≤ 0.05) lower at all of the evaluation dates (Figure 5c,d).

### 3.4. Peel Blemishes

Beit Alpha-type cucumbers have a smooth peel. However, after harvest, the fruit may suffer from water loss-related shriveling and develop physiological peel blemishes, such as pitting and warts. We found that the control, non-packed fruit suffered from more shriveling than the packed fruit. Under shelf conditions, the severity of shriveling among the control fruits was slight after 5 days, slight to moderate after 10 days and severe after 15 days. All of the bagged fruits showed significantly (*p* ≤ 0.05) lower values, and had only slight shriveling (Figure 6a). In the simulated farm-to-fork storage regime, barely any shriveling was observed immediately after marketing (2 days at 15 °C + 2 days at 22 °C). However, after 1 and 2 weeks of subsequent home refrigerated storage, the control fruit exhibited slight and moderate shriveling, respectively, while all of the packed fruits showed significantly (*p* ≤ 0.05) lower shriveling symptoms (Figure 6b).

Regarding pitting damage, only control fruits and fruits packaged in the non-perforated compostable packages showed slight but significantly (*p* ≤ 0.05) damage after 5 and 10 days under shelf conditions. However, after 15 days under shelf conditions, the control fruit suffered from severe pitting, while the fruit in the non-perforated and macro-perforated packages showed only slight to moderate damage. Fruits in the micro-perforated compostable packaging had the lowest pitting index (Figure 6c). In the simulated farm-to-fork supply-chain regime, all of the fruits had very low pitting damage after been exposed to the simulated marketing conditions. However, after 1 week of refrigerated home storage, the control fruits and the fruits packed in the non-perforated compostable bags had significantly (*p* ≤ 0.05) higher pitting damage, while there was only minimal damage among the fruits stored in the micro- and macro-perforated packages. After 2 weeks of home storage, the pitting damage was severe in the control fruit and moderate in the cucumbers kept in non-perforated packages. All of the fruits kept in micro- and macro-perforated packages showed very little pitting damage (Figure 6d).

Regarding the appearance of warts, after 10 and 15 days of the extended shelf-life regime (22 °C), the control, non-packed fruit and the fruit packaged in macro-perforated packages had significantly (*p* ≤ 0.05) more warts, whereas the fruits in the non-perforated and micro-perforated compostable packages barely had any warts at all (Figure 6e). Similar trends were observed under the simulated farm-to-fork supply-chain regime, in which the development of a warty appearance was prevented by keeping the cucumbers under modified atmosphere conditions in non-perforated or micro-perforated packages (Figure 6f).

### 3.5. Peel Color

The cucumbers were dark green at harvest, but tended to yellow during storage. Under continuous shelf conditions (at 22 °C), the control, non-packed fruit turned light green–yellow after 10 days and yellow–brown after 15 days (Figure 2 and Figure 7a). In contrast, all of the bagged fruits remained green after 10 days; but after 15 days, the green peel color was retained only among the cucumbers in the non-perforated and micro-perforated compostable packages (Figure 2 and Figure 7a). In the farm-to-fork supply-chain simulation, all of the fruits remained green after the simulated marketing and one additional week of refrigerated home storage. However, after 2 weeks of home storage, the control fruits as well as the fruits in the macro-perforated packages had turned light green; whereas the fruits in the sealed and micro-perforated packages remained mostly dark green (Figure 3 and Figure 7b).

### 3.6. Decay Development

After 10 days under continuous shelf conditions (22 °C), the fruit in the non-perforated packages showed a significantly (*p* ≤ 0.05) increase of up to 75% decay and the incidence of decay in these packages reached 100% after 15 days of shelf conditions. At the same time, the control fruits and the fruits in the macro-perforated packages developed between 40 to 70% decay, while the fruit in the micro-perforated packages did not show any decay at all (Figure 7c). In the farm-to-fork supply-chain regime, there was no decay after the marketing simulation. However, after one additional week of refrigerated home storage, the control, non-bagged fruit showed 12.5% decay. After 2 weeks of refrigerated home storage, the level of decay among the control fruits and the fruits in the non-perforated packages reached 37.5%, while there was a negligible incidence of decay among the fruits in the micro-perforated and macro-perforated packaging (Figure 7d).

### 3.7. Fruit Flavor

The flavor-acceptance score was 7.5 at harvest (on a scale of 1 to 9) and decreased during storage. During the continuous shelf-conditions (22 °C), the flavor-acceptance score of the control fruit gradually decreased to 6.0 after 5 days and to 5.0 after 10 days. After 15 days, the fruit was inedible (flavor score below 5; Figure 7e). The flavor scores of the fruits in the sealed packages were significantly (*p* ≤ 0.05) lower and were inedible (flavor score below 5) already after 5 days. The cucumbers stored in the micro-perforated and macro-perforated packages retained their flavor and had an acceptable flavor score (between 6.75 and 7.0) after 5 days and an edible flavor score (between 6.0 and 6.5) after 10 days, but were judged inedible after 15 days (Figure 7e).

At the end of the farm-to-fork chain simulation (i.e., after 2 weeks of home storage) the flavor score of the control, non-packed fruits had decreased from the initial score of 7.5 to just 5.5. As observed under shelf-conditions, the flavor scores of the fruits in the non-perforated packages were significantly (*p* ≤ 0.05) lower and were inedible (score of 2) due to a strong off-flavor. The flavor scores of the cucumbers in the micro-perforated and macro-perforated packages were slightly higher (between 6.0 and 7.0), but not significantly differ from that of the control fruit (Figure 7f).

### 3.8. Visual-Acceptance Score

The visual-acceptance score at harvest was 4.5 (i.e., between good and excellent). As can be seen in Figure 8a, during continuous storage under shelf conditions (22 °C), the control, non-bagged fruit remained acceptable after 5 days (visual-acceptance score of 3.2). However, after 10 days of shelf life, the visual-acceptance score of the control fruits was significantly (*p* ≤ 0.05) lower and declined far below the threshold value. Fruits in the non-perforated packages were also significantly (*p* ≤ 0.05) lower and unacceptable after 10 days, primarily due to the high incidence of decay. On the other hand, the quality of cucumbers in all of the micro- and macro-perforated packages remained well above the acceptance threshold after 10 days under shelf conditions (visual-acceptance scores of 3.1 to 3.8). Furthermore, the appearance of the fruits in the micro-perforated compostable packaging was acceptable even after 15 days of shelf conditions (Figure 8a).

In the farm-to-fork supply-chain regime, the appearance of all of the fruits remained acceptable after the simulated distribution-and-marketing. However, the visual-acceptance scores of the fruits packed in micro-perforated and macro-perforated packages were somewhat higher (between 4.0 and 4.25) than the scores of the control fruits and those of the fruits in non-perforated packages (between 3.2 and 3.3; Figure 8b). After one week of refrigerated home storage, the appearance of the control non-packed fruits was unacceptable (visual-acceptance score of 2.25). In contrast, fruits in the non-perforated packages were just above the minimum acceptability threshold (visual-acceptance score of 2.6), fruits in macro-perforated packages had moderate scores of 3.2 to 3.5, and fruits in the micro-perforated compostable packages had the highest visual-acceptance score of 4.0 (Figure 8b). After 2 weeks of home storage, only the fruits in the perforated packages remained acceptable, while the acceptance scores of the control fruit and of the fruit in the non-perforated packages were significantly (*p* ≤ 0.05) lower and below the acceptability limit (Figure 8b). It is worth noting that the visual-acceptance scores of the cucumbers stored in micro- and macro-perforated compostable packages were somewhat higher (3.25) than those of the cucumbers stored in the macro-perforated polypropylene bags (2.75), but that difference was not statistically significant.

## 4. Discussion

The main goals of the current study were to evaluate whether compostable packages may replace the conventional plastic films currently used to preserve the freshness and postharvest quality of cucumber fruits, and to get an idea of beneficial compostable packaging design for cucumbers. Cucumbers are highly perishable commodities prone to postharvest deterioration, caused by desiccation, senescence, and microbial spoilage. Efficient packaging should provide optimal storage microenvironment reducing moisture loss by enhanced air humidity, and inhibiting pathogen development and ethylene-induced senescence by elevated carbon dioxide and reduced oxygen levels. On the other hand, the extreme humidity and atmosphere composition changes should be avoided in order to prevent undesirable phenomena such as water condensation encouraging disease development, carbon dioxide injury and hypoxic fermentation causing off-flavors and tissue damage. For cucumbers, exposure to carbon dioxide levels above 5 [22,23] or 10% [24] results in injury manifested as enhanced decay susceptibility and yellowing. The low-oxygen tolerance of cucumbers is 3% [24].

The creation of modified atmosphere and humidity within fruit and vegetable packaging depends on the interaction between produce respiration and transpiration processes, and barrier properties of the packaging material towards gases and water vapor [25]. As compared to regular petroleum-based polypropylene film of the same thickness, the compostable packaging material examined in this study had five-fold higher water vapor transmission rate, but 2.5-fold lower oxygen permeability. Such trend is typical for biobased compostable films [26,27]. The low oxygen transmission rate of the compostable film increases the risk of hypoxic fermentation in respiring fresh produce like cucumbers. Such risk is especially high at super-optimal shelf-life temperature when enhanced respiratory consumption of oxygen is not balanced by adequate gas exchange through the packaging material.

Micro-perforation is an efficient way to prevent the hypoxia by matching the package oxygen permeability with produce requirements [28]. Perforations have great effect on oxygen and carbon dioxide transfer through packaging material, but much smaller influence on water vapor transmission that partially passes through polymer matrix [29]. This disparity allows the approach of modified atmosphere and modified humidity (MA–MH) packaging when desirable atmosphere composition is reached due to appropriate micro-perforation level, while in-package humidity is adjusted and water condensation is prevented by choosing a suitable plastic basis. The MA–MH approach was realized commercially in a series of micro-perforated packages based on polyamide blends having higher WVTR as compared to regular polyolefins [30].). The enhanced-WVTR compostable film demonstrated similar MA–MH potential, in addition to its sustainability advantage. Indeed, no water condensation was observed in the micro-perforated compostable packages, in contrast to the commercial polypropylene packs where profound accumulation of condensed water eventually resulted in sharp decay increase (Figure 7c). On the other hand, storage in micro-perforated compostable bags significantly reduced the weight loss of cucumbers as compared to the open containers, and prevented their shriveling (Figure 5a,b and Figure 6a,b). Future improvement of the compostable film formulation might allow modulating the WVTR value in order to reach even lower produce weight loss without compromising the condensation control. Mathematical modeling can be helpful for determining an optimal package micro-perforation level (28). However, such optimization has been out of the scope of the present study. The existing models typically based on conventional petroleum-based plastics need adjustment in order to allow reliable description of packaging systems involving relatively hydrophilic biodegradable materials. For example, in contrast to regular polyolefins the barrier properties of such films may be strongly affected by air humidity (Hong and Krochta, 2006) [31]. Moreover, in this work we have demonstrated for the first time the effect of biological factors such as mold development, on the barrier properties of biodegradable package (see below). Therefore, theoretical description of biodegradable packaging systems needs a special in-depth analysis that may be a subject of a separate study.

The results of this study clearly indicate that compostable packaging may preserve cucumber quality under various storage conditions: continuous shelf life at 22 °C and a simulated farm-to-fork supply chain, including distribution, marketing, and refrigerated home storage. The appropriate compostable packaging controlled fruit weight loss, shriveling, peel disorders, yellowing, decay, and flavor deterioration. However, the ability of the compostable packaging to preserve cucumber quality greatly depended on the perforation rate.

According to our results, the micro-perforated compostable packaging resulting in the creation of a modified atmosphere containing approximately 15–17% O_2_ and 2–5% CO_2_ was preferable for preserving cucumber quality, as compared to both non-perforated and macro-perforated packaging (Figure 4). This finding is in agreement with the previous observation of Manjunatha and Anurag [21], who reported that modified atmospheres containing 12–17% O_2_ and 5–10% CO_2_ have a positive effect on cucumber preservation. On the other hand, the macro-perforated packages had internal atmospheres similar to regular air and were somewhat less effective in reducing pitting, warts, yellowing, and decay development, as compared with the micro-perforated packages (Figure 6 and Figure 7).

We found that storage of cucumbers in non-perforated compostable packages resulted in the development of hypoxic conditions with just ~1.5% O_2_ (Figure 4), which is below the tolerance level of cucumbers [19,24]. Accordingly, using sealed compostable packages enhanced decay development and harmed fruit flavor. Therefore, non-perforated packages are not recommended for the storage and marketing of cucumbers. The sharp increase in the level of oxygen in the atmosphere within the non-perforated packages during the second half of the storage period (Figure 4a,b) requires further examination. During the trials, we observed the effect of spoilage microorganisms on the integrity of the compostable films (Figure 9). Once this air influx into the non-perforated packages coincided with the onset of profound decay on hypoxia-damaged cucumbers, it was probably associated with partial microbial degradation of the packaging. However, such re-exposure to oxygen did not reverse the hypoxic damage to plant tissues and might have even aggravated that damage [32].

We further compared the effects of the compostable packaging with those of commercial macro-perforated polypropylene plastic packaging and found that the compostable packaging reduced cucumber weight loss almost as well as the polypropylene plastic packaging, but without the creation of condensation (Figure 5). Low condensation rates constitute an important advantage of the compostable packages over regular plastic films, since the accumulation of free water within the packages may enhance microbial spoilage and harm food safety [33]. The low-condensation characteristics of the compostable packages also eliminate the need to add absorbing pads, which make the packaging process more complex and expensive [34]. Nonetheless, it should be mentioned that we hereby examined a specific type of a compostable packaging material manufactured by TIPA^®^ Corp., while other types of compostable materials may have different water vapor permeability’s and consequent condensation rates [35]. Furthermore, various additives that may be applied to the compostable films, such as soybean oil, may further influence water resistance properties [36].

Other advantages of the micro-perforated compostable packages over the macro-perforated polypropylene packages are the observed reductions in warts, decay, and yellowing (Figure 6e,f and Figure 7a,c). Nonetheless, these effects are probably related to the creation of favorable modified atmospheres within the micro-perforated packages rather than any unique properties of the compostable film.

## 5. Conclusions

Overall, our results indicate that, compared to non-packed produce, the micro-perforated compostable packaging extended the shelf-life of cucumbers kept at 22 °C by at least 5 days and increased cucumber shelf-life by more than 7 days in a simulated supply-chain treatment that was followed by refrigerated home storage (Figure 8). These results regarding the extension of the postharvest storage life of cucumbers by compostable packages are in agreement with previous reports demonstrating the advantages of MAP for extending the storage life of cucumbers [20,21,37]. Nevertheless, here, we have demonstrated for the first time that compostable packaging material may provide an environmentally friendly alternative to the commonly used plastic retail packaging [38].

## Figures and Tables

**Figure 1 foods-10-00471-f001:**
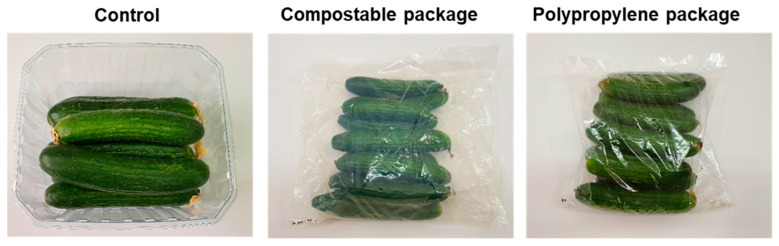
Visual appearance of control, compostable and polypropylene packages of cucumber fruit.

**Figure 2 foods-10-00471-f002:**
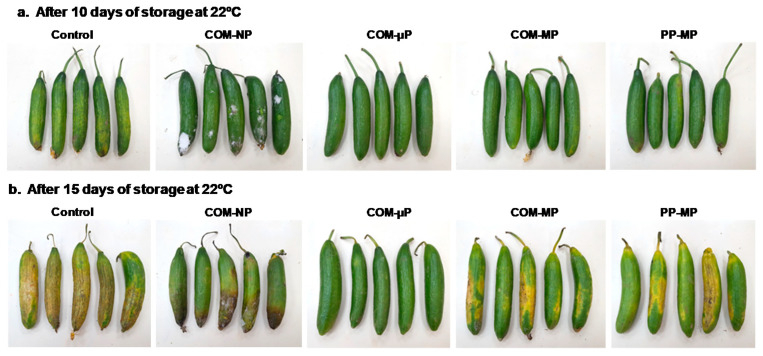
Photographs of cucumber fruits stored in different types of packaging for (**a**) 10 days or (**b**) 15 days under continuous shelf conditions at 22 °C. COM—compostable, PP—polypropylene, µ—micro-perforated, M—macro-perforated, NP—non-perforated.

**Figure 3 foods-10-00471-f003:**
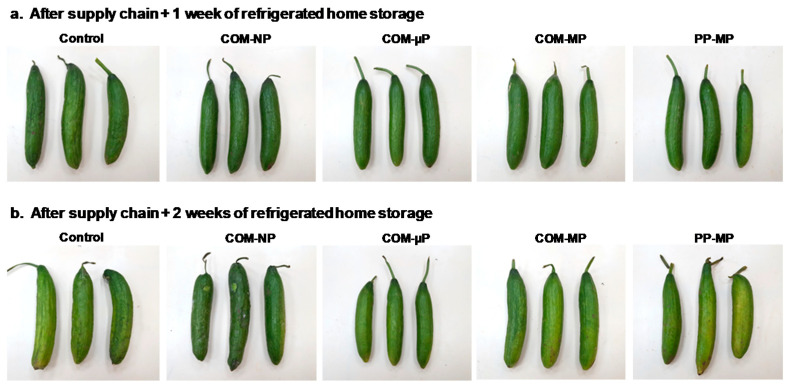
Photographs of cucumber fruits stored in different types of packaging under simulated farm-to-fork supply-chain conditions including distribution and marketing (2 days at 15 °C + 2 days at 22 °C) plus (**a**) 1 week or (**b**) 2 weeks of refrigerated home storage at 4 °C. COM—compostable, PP—polypropylene, µP—micro-perforated, MP—macro-perforated, NP—non-perforated.

**Figure 4 foods-10-00471-f004:**
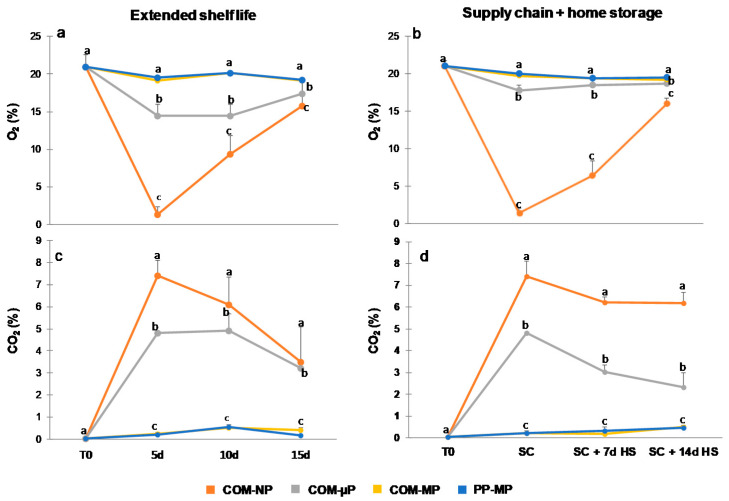
Oxygen and carbon dioxide levels in the headspaces of different cucumber packages. Measurements were taken during continuous storage under shelf conditions at 22 °C (Extended shelf life) and after the simulated farm-to-fork supply-chain treatment (2 days at 15 °C + 2 days at 22 °C) plus 1 or 2 weeks of refrigerated home storage at 4 °C (Supply chain + home storage). Figures (**a**) and (**b**) represent O_2_ levels, and Figures (**c**) and (**d**) represent CO_2_ levels after extended shelf life and farm-to-fork supply-chain conditions, respectively. Data are means ± SE of 3 replications. Different letters indicate significant differences at *p* ≤ 0.05. COM—compostable, PP—polypropylene, µP—micro-perforated, MP—macro-perforated, NP—non-perforated.

**Figure 5 foods-10-00471-f005:**
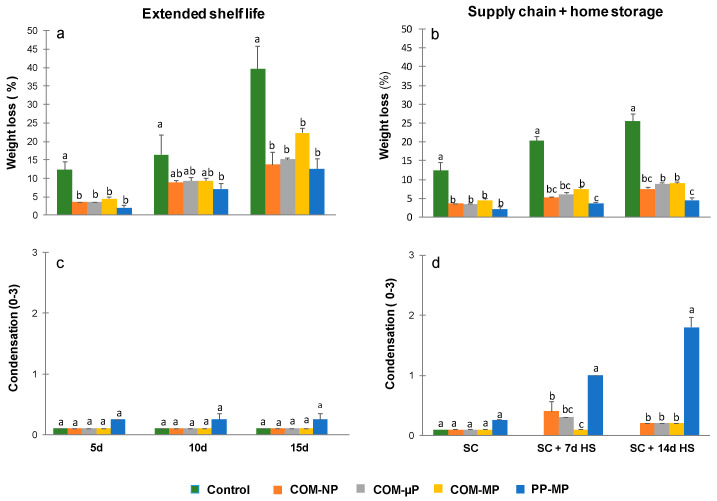
Weight loss and condensation levels in different cucumber packages. Measurements were taken during continuous storage under shelf conditions at 22 °C (Extended shelf life) and after the simulated farm-to-fork supply-chain treatment (2 days at 15 °C + 2 days at 22 °C) plus 1 or 2 weeks of home refrigerated storage at 4 °C (Supply chain + home storage). Figures (**a**) and (**b**) represent weight loss levels, and Figures (**c**) and (**d**) represent condensation levels after extended shelf life and farm-to-fork supply-chain conditions, respectively. Data are means ± SE of 3 replications. Different letters indicate significant differences at *p* ≤ 0.05. COM—compostable, PP—polypropylene, µP—micro-perforated, MP—macro-perforated, NP—non-perforated.

**Figure 6 foods-10-00471-f006:**
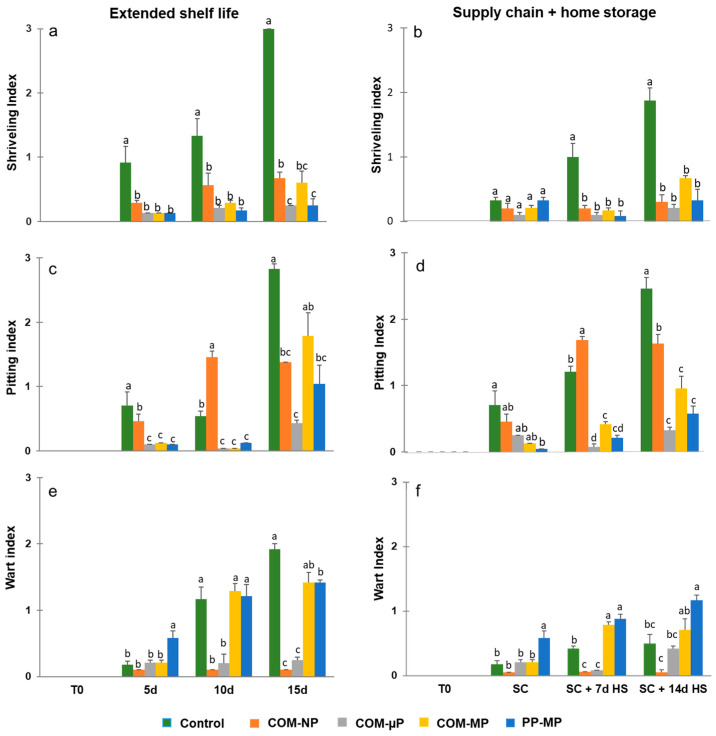
Shriveling, pitting, and wart indices of cucumbers stored in different types of packaging. Measurements were taken during continuous storage under shelf conditions at 22 °C (Extended shelf life), and after the simulated farm-to-fork supply-chain treatment (2 days at 15 °C + 2 days at 22 °C) plus 1 or 2 weeks of refrigerated home storage at 4 °C (Supply chain + home storage). Figures (**a**) and (**b**) represent shriveling indices; Figures (**c**) and (**d**) represent pitting indices, and Figures (**e**) and (**f**) represent wart indices after extended shelf life and farm-to-fork supply-chain conditions, respectively. Data are means ± SE of 3 replications. Different letters indicate significant differences at *p* ≤ 0.05. COM—compostable, PP—polypropylene, µP—micro-perforated, MP—macro-perforated, NP—non-perforated.

**Figure 7 foods-10-00471-f007:**
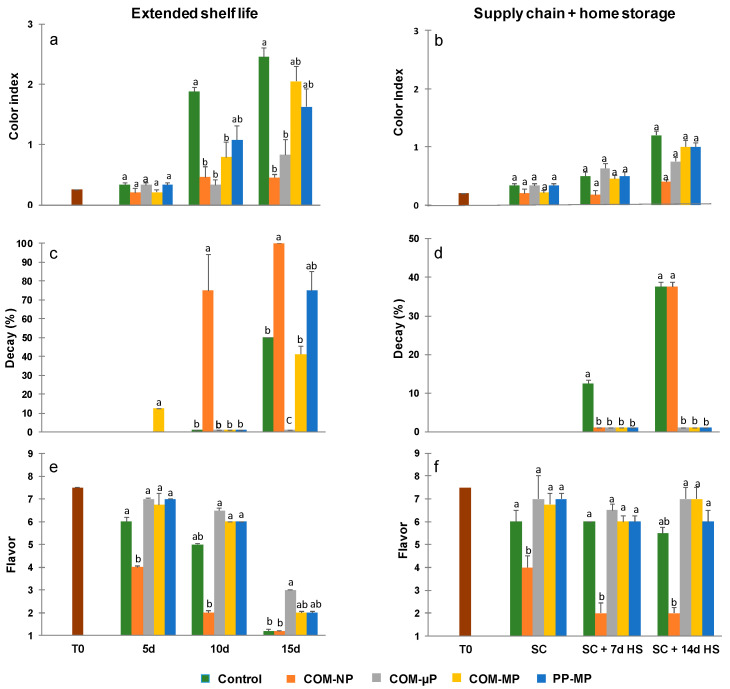
Color indices, decay levels, and flavor scores of cucumbers kept in different types of packaging. Measurements were taken during continuous storage under shelf conditions at 22 °C (Extended shelf life) and after the simulated farm-to-fork supply-chain treatment (2 days at 15 °C + 2 days at 22 °C) plus 1 or 2 weeks of refrigerated home storage at 4 °C (Supply chain + home storage). Figures (**a**) and (**b**) represent color indices; Figures (**c**) and (**d**) represent decay precentages, and Figures (**e**) and (**f**) represent flavor acceptance scores after extended shelf life and farm-to-fork supply-chain conditions, respectively. Data are means ± SE of 3 replications. Different letters indicate significant differences at *p* ≤ 0.05. COM—compostable, PP—polypropylene, µP—micro-perforated, MP—macro-perforated, NP—non-perforated.

**Figure 8 foods-10-00471-f008:**
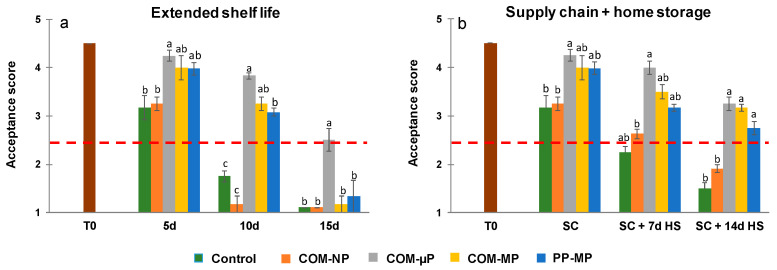
Acceptance scores of cucumbers kept in the different types of packaging. Measurements were taken during continuous storage under shelf conditions at 22 °C (Extended shelf life) and after the simulated farm-to-fork supply-chain treatment (2 days at 15 °C + 2 days at 22 °C) plus 1 or 2 weeks of refrigerated home storage at 4 °C (Supply chain + home storage). Figures (**a**) and (**b**) represent overall acceptance scores after extended shelf life and farm-to-fork supply-chain conditions, respectively. Data are means ± SE of 3 replications. The red dashed line indicates the minimum acceptance threshold of 2.5. Different letters indicate significant differences at *p* ≤ 0.05. COM—compostable, PP—polypropylene, µP—micro-perforated, MP—macro-perforated, NP—non-perforated.

**Figure 9 foods-10-00471-f009:**
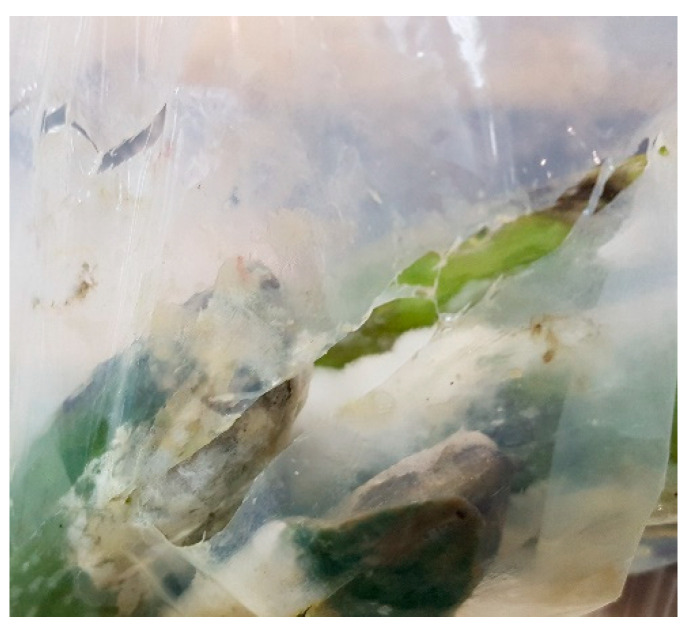
Effects of spolage organisims on the integrity of compostable packages.

## Data Availability

Not applicable.

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
