# Peer review of "Effects of Compostable Packaging and Perforation Rates on Cucumber Quality during Extended Shelf Life and Simulated Farm-to-Fork Supply-Chain Conditions"

_foods, 2021, doi:10.3390/foods10020471_

Round 1

Reviewer 1 Report

I have the following comments and suggestions:

  1. FAO abbreviation is not explained.
  2. Place figures under the belonging text.
  3. Figure 5, 6, 7, 8: the color of control samples is similar to other samples’ marking.
  4. Figure 5, 6, 7, 8: standard deviation is no fully visible.
  5. Figure 4: standard deviation is no fully visible.
  6. Discussion part should be prolonged; it is too short, more references are need it.
  7. The following reference should be used, since it is describing the possibility of biodegradable packaging: Jamróz, E., Kopel, P., Tkaczewska, J., Dordevic, D., Jancikova, S., Kulawik, P., ... & Adam, V. (2019). Nanocomposite Furcellaran Films—The Influence of Nanofillers on Functional Properties of Furcellaran Films and Effect on Linseed Oil Preservation. Polymers11(12), 2046.

Author Response

FAO abbreviation is not explained – we  have now indicated the full name of FAO

Place figures under the belonging text - Thanks for the comment. The Figures are now placed under the belonging text.

Figure 5, 6, 7, 8: the color of control samples is similar to other samples’ marking – Thanks for the comment. We have now changed the color of the legend of the control sample in order to better distinguish it from the other treatments.

Figure 5, 6, 7, 8: standard deviation is no fully visible - Thanks for the comment. We now have ensured that the standard deviations are clear and visible.

Figure 4: standard deviation is no fully visible - Thanks for the comment. We now have ensured that the standard deviations are clear and visible.

Discussion part should be prolonged; it is too short, more references are need it – we have significantly extended the Discussion section , and have added several new paragraphs, eight new references, and a new figure. 

The following reference should be used, since it is describing the possibility of biodegradable packaging: Jamróz, E., Kopel, P., Tkaczewska, J., Dordevic, D., Jancikova, S., Kulawik, P., ... & Adam, V. (2019). Nanocomposite Furcellaran Films— The Influence of Nanofillers on Functional Properties of Furcellaran Films and Effect on Linseed Oil Preservation. Polymers11(12), 2046 – We have now cited the suggested reference (see ref. 18).

Reviewer 2 Report

Dear Editor,

This study deals with the effect of compostable packages with different perforation rate on the quality of cucumber over time under different storage conditions.

The topic is basic but novel considering the existing literature. The manuscript is well structured. The Introduction section gives a logical account of the current state of the technology for packaging and preserving cucumber and provides a justification for the current work. The experiment design and interpretation of the results are sound. The data are presented properly, and the study objective is justified. Considering experimental results and practical application of this study for preserving highly perishable cucumber and the novelty of the experiment.

- There are minor English grammar errors throughout the manuscript.

- To increase the readability of the article and to have a deeper understanding of the research content for readers, please add more details in the discussion to support your research results.

- Please respect subscript in writing CO2 and O2 in the whole manuscript.

- It would be more precise to add inside the text if the different treatments had significant differences or not or even by simply by adding p<0.05 or p>0.05.

- Line 122-132: Please delete this author guideline paragraph.

- Figures: Please tick mark the Y-axis numbers.

Author Response

The topic is basic but novel considering the existing literature. The manuscript is well structured. The Introduction section gives a logical account of the current state of the technology for packaging and preserving cucumber and provides a justification for the current work. The experiment design and interpretation of the results are sound. The data are presented properly, and the study objective is justified. Considering experimental results and practical application of this study for preserving highly perishable cucumber and the novelty of the experiment – Thank you for the positive impression.

There are minor English grammar errors throughout the manuscript – We carefully revised the English grammar throughout the manuscript. The manuscript was also revised by a professional English language editor.

To increase the readability of the article and to have a deeper understanding of the research content for readers, please add more details in the discussion to support your research results – We have now significantly deepened the Discussion chapter, including the addition of several new paragraphs, citation of new references, and addition of a new Figure (a photograph demonstrating the effects spoilage microrganisims on the integrity of the compostable packages).  We also further discussed  the factors affecting the accumulation of modified atmospheres and modified humidity (MA/MH) in the packages.

Please respect subscript in writing CO2 and Oin the whole manuscript – Thank you for the comment. We now have ensured that all indications of O2 and CO2 are written in subscript.

It would be more precise to add inside the text if the different treatments had significant differences or not or even by simply by adding p<0.05 or p>0.05 – we have now indicated significant differences throughout the text in the Results chapter.

Line 122-132: Please delete this author guideline paragraph –  Thank you for drawing our attention to this template fragment erroneously left in the text. We have now deleted it.

Figures: Please tick mark the Y-axis numbers – Thank you for your comment. We now added tick marks to the Y-axis of all figures.

Reviewer 3 Report

Dear authors,

the results show that postharvest storage life of cucumbers can be enhanced by using compostable packages. This result is not surprising but supports the application of biomaterial.

General comment:

  • You rightly stated that the main issue is the perforation of the film and should have a closer view on this issue: In your article, you use a descriptive approach. You should support and enhance the results and conclusion by discussion of the mass transport properties of micro perforated films against the respiration rate of cucumbers. When you compare the mass transport with the respiration rate you can evaluate the content of oxygen and carbon dioxide in the head space and then compare these values with your measured values. You can use eg. the following articles that review the determination and modelling of O2 and CO2 transmission rates through microperforated films (J. Gonza´ lez et al. / Journal of Food Engineering 86 (2008) 194–201 and S.C. Fonseca et al. / Journal of Food Engineering 52 (2002) 99–119). But you will surely find other articles you can cite.

Line 94/95: Please specify the measurement conditions (temperature and rel. humidity?)

Figure 4 ff: Please explain the exact meaning of a, b, c in the figures.

Author Response

The results show that postharvest storage life of cucumbers can be enhanced by using compostable packages. This result is not surprising but supports the application of biomaterial - Thanks for your supporting remark.

General comment:

You rightly stated that the main issue is the perforation of the film and should have a closer view on this issue: In your article, you use a descriptive approach. You should support and enhance the results and conclusion by discussion of the mass transport properties of micro perforated films against the respiration rate of cucumbers. When you compare the mass transport with the respiration rate you can evaluate the content of oxygen and carbon dioxide in the head space and then compare these values with your measured values. You can use eg. the following articles that review the determination and modelling of O2 and CO2 transmission rates through microperforated films (J. Gonza´ lez et al. / Journal of Food Engineering 86 (2008) 194–201 and S.C. Fonseca et al. / Journal of Food Engineering 52 (2002) 99–119). But you will surely find other articles you can cite – We significantly extended the Discussion section, and have added several new paragraphs, references and an additional Figure. As requested, we now specifically discussed  the factors affecting the modified atmospheres and modified humidity (MA/MH) accumulation and mass transport through the packages, and have cited the suggested references.

Line 94/95: Please specify the measurement conditions (temperature and rel. humidity?) – We now have indicated in the M&M, section 2.3, the detailed temperature and relative humidity conditions in all storage conditions.

Figure 4 ff: Please explain the exact meaning of a, b, c in the figures – It is mentioned in the legend of Fig. 4 that Different letters indicate significant differences at p ≤ 0.05.

Round 2

Reviewer 1 Report

The article can be accepted now.